# Controlled Crystallinity of a Sn-Doped α-Ga_2_O_3_ Epilayer Using Rapidly Annealed Double Buffer Layers

**DOI:** 10.3390/nano14020178

**Published:** 2024-01-12

**Authors:** Kyoung-Ho Kim, Yun-Ji Shin, Seong-Min Jeong, Heesoo Lee, Si-Young Bae

**Affiliations:** 1Semiconductor Materials Center, Korea Institute of Ceramic Engineering and Technology, Jinju 52851, Republic of Korea; energykkh@gmail.com (K.-H.K.); shinyj@kicet.re.kr (Y.-J.S.); smjeong@kicet.re.kr (S.-M.J.); 2School of Materials Science and Engineering, Pusan National University, Busan 46241, Republic of Korea; heesoo@pusan.ac.kr

**Keywords:** Ga_2_O_3_, mist CVD, doping, buffer layer, mobility

## Abstract

Double buffer layers composed of (Al_x_Ga_1−x_)_2_O_3_/Ga_2_O_3_ structures were employed to grow a Sn-doped α-Ga_2_O_3_ epitaxial thin film on a sapphire substrate using mist chemical vapor deposition. The insertion of double buffer layers improved the crystal quality of the upper-grown Sn-doped α-Ga_2_O_3_ thin films by blocking dislocation generated by the substrates. Rapid thermal annealing was conducted for the double buffer layers at phase transition temperatures of 700–800 °C. The slight mixing of κ and β phases further improved the crystallinity of the grown Sn-Ga_2_O_3_ thin film through local lateral overgrowth. The electron mobility of the Sn-Ga_2_O_3_ thin films was also significantly improved due to the smoothened interface and the diffusion of Al. Therefore, rapid thermal annealing with the double buffer layer proved advantageous in achieving strong electrical properties for Ga_2_O_3_ semiconductor devices within a shorter processing time.

## 1. Introduction

Gallium oxide (Ga_2_O_3_) stands out as a next-generation high-voltage power semiconductor material with a wide bandgap (4.6–5.3 eV) and a high Baliga figure of merit (~3444) compared to that of Si [1,2,3,4,5,6]. The crystal structure of Ga_2_O_3_ is known to exist in six crystal phases (α, β, δ, γ, ε, and κ) [7]. The most thermally stable phase among them (the monoclinic β structure) has been exclusively grown as a single crystal bulk at a melting temperature of ~1800 [8]. The rhombohedral α-Ga_2_O_3_, while thermodynamically metastable, boasts the widest bandgap (5.3 eV) and a high breakdown voltage relative to other Ga_2_O_3_ crystal phases [9,10]. Despite the lattice constant differences of approximately 3.3% and 4.8% in the *c*- and *a*-axes, heteroepitaxial α-Ga_2_O_3_ thin films have been successfully grown on sapphire (α-Al_2_O_3_) substrates [11], leading to the generation of misfit dislocation in a period of 8.6 nm along the [1$\bar{1}$00] direction at the interface between the α-Ga_2_O_3_ epilayer and the sapphire substrate [12,13].

The heteroepitaxy of the corundum structure system facilitates the growth of various solid solutions with transition metal ions (α-M_2_O_3_; M = In, Al, Fe, Cr, V, Ti, Rh, and Ir) [13]. In particular, ternary composition structures such as α-(Al_x_Ga_1−x_)_2_O_3_ and α-(In _x_Ga_1−x_)_2_O_3_ have been studied [14,15,16,17]. The ternary composition layer of α-(Al_x_Ga_1−x_)_2_O_3_ widens the bandgap, enhances thermal stability, and reduces interfacial strain between the grown Ga_2_O_3_ epilayer and sapphire substrates, albeit with a slower growth rate. On the other hand, α-(In _x_Ga_1−x_)_2_O_3_ allows for a narrowing of the bandgap with a relatively high growth rate. Hence, adjusting the composition fraction of the crystal structure offers possibilities for both widening and narrowing the bandgap. These ternary composition layers can be employed to relax the interfacial strain in the heteroepitaxial layer of the corundum structure system [15].

Extrinsic doping of heteroepitaxial α-Ga_2_O_3_ thin films has facilitated advancements in high-power semiconductor applications. While an unintentionally doped α-Ga_2_O_3_ epilayer exhibited insulator-like characteristics of high resistivity (>1 × 10^5^ Ω∙cm) [18], a doped α-Ga_2_O_3_ epilayer featured high conductivity with n-type dopants such as Si, Sn, and Ge [19,20,21]. The ionic radius of Sn (0.069 nm) is similar to that of Ga (0.062 nm), allowing easy substitution for Ga without serious lattice distortion [22]. In both α- and β-Ga_2_O_3_ thin films, Sn doping has effectively controlled the carrier concentration in the range of 10^17^ to 10^19^ cm^−3^ [23]. Notably, the experimentally measured electron mobility of β-Ga_2_O_3_ (<140 cm^2^/V·s) was much higher than that of α-Ga_2_O_3_ thin films (<24 cm^2^/V·s) [20,24]. The limitation of electron mobility is mainly attributed to the high density of dislocations in heteroepitaxial α-Ga_2_O_3_ thin films [24]. Dislocation scattering can be stronger due to charged free carriers that are trapped with Coulomb force, decreasing electron mobility in heteroepitaxial α-Ga_2_O_3_ thin films compared to that in homoepitaxial β-Ga_2_O_3_ thin films [24]. Several approaches have been suggested to suppress the dislocation density of α-Ga_2_O_3_ thin films, including lateral overgrowth, solid-solution buffer layers, and heat treatment [25,26,27,28,29]. Lateral overgrowth induced the bending of dislocations in regrown Ga_2_O_3_ thin films by limiting the growth region with a patterned dielectric mask, achieving a high crystal quality in Ga_2_O_3_ epilayers [25]. A buffer layer with a ternary solid solution was also demonstrated with quasi-graded α-(Al, Ga)_2_O_3_ buffer layers [29]. It decreased the lattice mismatch between α-Ga_2_O_3_ and α-Al_2_O_3_ to be ~1% on the *a*-axis, reducing the generation of dislocations by the substrate [29]. Akaiwa et al. achieved an electron mobility of 24 cm^2^/V·s in Sn-doped α-Ga_2_O_3_ thin films on sapphire substrates using a low temperature (500 °C) for long-term annealing (24 h), which induced highly resistive characteristics in the underlying layer and blocked the edge dislocation transferred from the substrate [24].

Several epitaxial methods for growing heteroepitaxial α-Ga_2_O_3_ thin films have been studied, including halide vapor phase epitaxy (HVPE), metalorganic chemical vapor deposition (MOCVD), molecular beam epitaxy (MBE), pulsed laser deposition (PLD), and mist chemical vapor deposition (mist CVD) [12,25,30,31,32]. Among them, mist CVD provides a cost-effective solution for growing a ternary solid solution. In this study, we controlled the crystallinity of α-Ga_2_O_3_ thin films grown through mist CVD through double buffer layer combinations and rapid thermal annealing. Double buffer layers consisting of α-Ga_2_O_3_ and α-(Al_x_Ga_1−x_)_2_O_3_ layers were chosen for the strain relaxation of the heteroepitaxial interface. The temperature of rapid thermal annealing was adjusted to control the phase transition and interfacial quality of the grown layers over a short time. Afterward, a Sn-doped α-Ga_2_O_3_ epilayer was successfully grown with high crystallinity and electron mobility, serving as a template for next-generation power semiconductor devices.

## 2. Materials and Methods

Mist CVD was used to grow Sn-doped Ga_2_O_3_ thin films with double buffer layers. Figure 1 shows the three steps (A–C) of the procedure in this study. In step A, double buffer layers of α-Ga_2_O_3_ and α-(Al_x_Ga_1−x_)_2_O_3_ were grown on a *c*-plane sapphire substrate. In Step B, rapid thermal annealing was conducted for the buffer layers. In step C, a Sn-doped Ga_2_O_3_ thin film was grown on the (Al_x_Ga_1−x_)_2_O_3_/Ga_2_O_3_ buffer layers. Table 1 summarizes the experimental conditions in detail. As precursors for the mist CVD process, metal acetylacetonates (Me(acac)_3_, Alpa Aesar, Haverhill, MA, USA) were used. For the growth of the double buffer layers in step A, the α-Ga_2_O_3_ layer was grown with a 0.05-M Ga solution for 1 h, while (Al_x_Ga_1−x_)_2_O_3_ was grown by adjusting the 0.03-M Ga and 0.18-M Al precursor solutions for 2 h. Step C involved mixing a 0.1 mol% SnCl_2_ solution with a 0.05-M Ga precursor solution for 3 h to generate a Sn-Ga_2_O_3_ layer. An ultrasonic transducer generated mist with a frequency of 1.7 MHz. The solution temperature was maintained at 28–30 °C, with flow rates of 5 L/min for carrier gas (air) and dilution gas (O_2_). Inside the furnace, a *c*-plane sapphire substrate was loaded with a susceptor inclined at 45°. The growth temperature was maintained at 450–500 °C. The processing time for rapid thermal annealing in step B was 3 min. To measure the crystal structure of the grown layers, θ/2θ scans were performed using X-ray diffraction (XRD) with Cu–Kα1 (λ = 1.54056 Å) and Kα2 (λ = 1.54440 Å) (D8 ADVANCE, Bruker, Billerica, MA, USA). The surface structures were characterized by using field-emission scanning electron microscopy (FE-SEM) (MIRA3 LM, TESCAN, Brno, Czech Republic). An ultraviolet–visible (UV-Vis) spectrophotometer (Cary 5000, Agilent, Santa Clara, CA, USA) was used to measure transmittance, and the surface morphology was observed using an atomic force microscope (JSPM-5200, JEOL, Tokyo, Japan). Transmission electron microscopy (TEM) (JEM-ARM200F, JEOL, Tokyo, Japan) was applied for microstructural characterizations. The electrical properties of the grown Sn-Ga_2_O_3_ were characterized through Hall effect measurements (HMS-5000, Ecopia, Anyang, Republic of Korea), thereby obtaining the carrier concentration and mobility.

## 3. Results

### 3.1. Growth of (Al_x_Ga_1−x_)_2_O_3_/Ga_2_O_3_ Buffer Layers and Rapid Thermal Annealing

At first, double buffer layers of α-Ga_2_O_3_ and α-(Al_x_Ga_1−x_)_2_O_3_ were grown as sample A1 in step A. In step B, sample A1 underwent annealing at 700–800 °C (samples B2–B4) for 3 min. Figure 2 displays the top and cross-sectional views of FE-SEM images of the grown samples (A1, B2, B3, and B4) in Figure 2a–h, respectively. Triangular island-shaped grains were observed on the surfaces shown in Figure 2a–d. We speculate that the triangular grain formation resulted from the Stranski–Krastanov growth mode due to the chemical instability of group III species and the mixing of metastable crystal phases [33,34,35]. Additionally, these grains morphologically corresponded to the α phase, which was partially retained even after the annealing procedure in step B [36]. However, as shown in Figure 2d, line-shaped boundaries appeared on the top surface of the α-(Al_x_Ga_1−x_)_2_O_3_/α-Ga_2_O_3_ layers, which was possibly due to the formation of crystal grains originating from the $\left(\bar{4}02\right)$ planes of the β phase [22]. The thickness of the grown layers ranged from 278 to 365 nm for samples A1–B4, as shown in Figure 2e–h. Horizontal mist CVD was responsible for the slight variation in thickness, as it commonly results in varied growth rates depending on the sample position [37]. The α-(Al_x_Ga_1−x_)_2_O_3_ layer was barely identifiable in the SEM observation because of its thinness (less than 80 nm) and the phenomenon of partial electron charging [16].

Figure 3a presents an XRD θ/2θ scan of the grown α-(Al_x_Ga_1−x_)_2_O_3_/α-Ga_2_O_3_. The (006) peak corresponding to α-Ga_2_O_3_ was observed at 40.25°. The α-(Al_x_Ga_1−x_)_2_O_3_ layer’s peak almost merged with the α-Ga_2_O_3_ layer’s peak, confirming a compositional structure of α-(Al_0.3_Ga_0.7_)_2_O_3_ through the calculation of Vegard’s law [38]. The low Al contents might have been due to the incomplete decomposition of the Al precursor at the relatively low growth temperature of 500 °C [14]. This incomplete growth condition of Al species could also account for the formation of triangular grains. During rapid thermal annealing, β phases were observed at 38.3° above the annealing temperature of 750 °C (samples B3 and B4), and the (006) peak of α-Ga_2_O_3_ was almost diminished at 800 °C (sample B4), which suggested significant phase transitions from the α to the β phase at the annealing temperatures of 750 and 800 °C, respectively. The κ phase was slightly mixed in all samples, a common occurrence in α-Ga_2_O_3_ thin films grown on sapphire substrates, as observed in other literature [39]. Typically, α-Ga_2_O_3_ undergoes a phase transition to the β phase above 600 °C [40]. For the thin films grown in this study, the tolerance temperature of the phase transition was increased by inserting the α-(Al_0.3_Ga_0.7_)_2_O_3_ thin film. Notably, the phase transition temperature of Ga_2_O_3_ thin films could be shifted upward by thinning the thickness or increasing the Al content [41]. Figure 3b shows the full width at half maximum (FWHM) of the X-ray rocking curve (XRC) ω scan to confirm the crystal quality in processing steps A and B. In the XRD reflection of the (006) symmetry plane, the FWHMs of XRC were 57, 89, and 112 arcsec in A1, B2, and B3, respectively, indicating that the crystallinity in the (006) plane slightly deteriorated as the annealing temperature increased. On the other hand, in the XRD reflection of the (104) asymmetry plane, the FWHMs of the XRC were 2503, 2282, and 2382 arcsec for samples A1, B2, and B3, respectively. We speculate that the annealing temperature of 700 °C affected the re-crystallization of the grown thin films, thereby improving the crystallinity of the asymmetric components. Conversely, the higher annealing temperature resulted in poor crystallinity with the β-phase mixing. For sample B4, the β-phase components of the XRC were predominantly measured due to phase shift. The FWHMs of the XRC on the (−201) and (−402) reflections were 3034 and 3548 arcsec, respectively. Hence, the crystallinity of the phase-transition β-Ga_2_O_3_ was not as good as that of the pure single crystals [30].

Figure 4 presents the Tauc plots of the transmittances for samples A1–B4. It was fitted with Equation (1) [42].
(1)$$(\alpha h\nu )={B(h\nu -E_{g})}^{1/2}$$
where *α* is the absorption coefficient, *hν* is the energy of the incident photon, *B* is the absorption edge with a parameter, and *E*_g_ is the bandgap. The reference specimen, sample A1, showed a bandgap of 5.30 eV. The bandgaps of samples B2 and B3 were also ~5.30 eV. It was optically evident that the α-phase region was almost dominant in sample B3, although the β phase was partially included. The bandgap of sample B4 was 4.9 eV, matching the abrupt phase transition from the α phase to the β phase inside the α-(Al_0.3_Ga_0.7_)_2_O_3_/α-Ga_2_O_3_ structure well.

### 3.2. Growth of the Sn-Doped α-Ga_2_O_3_ Thin Film on the Double Buffer Layers

In step C, Sn-doped α-Ga_2_O_3_ epilayers (samples C1–C4) were grown on samples A1–B4, respectively. Figure 5 shows the top view (Figure 5a–d) and cross-sectional view (Figure 5e–h) of FE-SEM images for samples C1–C4. A high density of sub-micrometer-scale islands covered the entire surface of sample C1 (Figure 5a). However, a few micrometer-scale grains partially protruded on the surfaces of samples C2 and C3 (Figure 5b,c). Flake-like grains were exceptionally formed on the surface of sample C4 (Figure 5d). The inhomogeneous surface morphology of samples C1–C4 was related to phase mixing depending on the thermal annealing temperatures. The total thickness of the Sn-Ga_2_O_3_/(Al_0.3_Ga_0.7_)_2_O_3_/Ga_2_O_3_ layers was 2114, 2256, 2316, and 2306 nm for samples C1, C2, C3, and C4, respectively, as shown in Figure 5e–h. Despite the various crystal phases and qualities, the thickness variations were relatively small considering the variations in the growth rate in horizontal mist CVD. Unlike in the cross-sectional SEM images of samples A1–B4 in Figure 2e–h, a dark contrast corresponding to the (Al_0.3_Ga_0.7_)_2_O_3_ layer was observed between the Sn-Ga_2_O_3_ and Ga_2_O_3_ layers. The dark interfacial contrast gradually diminished with increasing annealing temperatures, as shown in Figure 5f–h, which was possibly due to the diffusion of Al during step B.

Afterward, a θ/2θ XRD scan was conducted to study the crystal structure of the grown Sn-doped α-Ga_2_O_3_ epilayers, as shown in Figure 6a. In all samples (C1–C4), the (006) peak of the Sn-doped α-Ga_2_O_3_ layer and the (006) peak of α-Al_2_O_3_ were observed. The relative intensity of the κ-phase peaks gradually increased from sample C1 to sample C4, indicating that the ratio of the κ-phase increased in the Sn-Ga_2_O_3_ epilayers. Notably, sample C4 included the β phase, while the β-phase peak of sample C3 was weak. Hence, the inclusion of the β phase could be suppressed in the topmost Ga_2_O_3_ epilayer by the growth competition between the α and β phases, whereas the κ phase almost succeeded in the upper layers. Figure 6b shows the FWHMs of the XRC for samples C1–C4. In the symmetric (006) plane, the values of 57, 96, 79, and 1814 arcsec were obtained for samples C1–C4. On the other hand, in the asymmetric (104) plane, values of 1611, 1549, 1651, and 3826 arcsec were confirmed for samples C1–C4, respectively. Note that the FWHM of (104) for sample C2 was lower than that for sample C1, while the FWHM of (006) for sample C2 was higher than that for sample C1, showing typical crystallinity in the lateral growth. The improved crystallinity might be attributed to the growth competition of the α and κ phases, thereby improving the crystal quality via local lateral growth. However, when the κ and β phases were mixed a lot, the crystal quality became poor, as shown in sample C4 (Figure 6b). An abrupt increase in the FWHM of the XRC was found in sample C4 with a high ratio of the β phase.

Figure 7 shows the distribution of the root-mean-square (RMS) roughness of all samples (A1–C4) measured with AFM, and the dashed line indicates the average of each sample. Sample A1 had the lowest RMS roughness, averaging 3.7 nm. The roughness of samples B2–B4 gradually increased to 5.6, 9.1, and 9.6 nm, respectively, as the annealing temperature increased. This implied that the alignment of crystal grains was distorted by the rapid thermal annealing process during step B. In the case of samples C1–C4, the average RMS roughness was 7.0, 26.0, 26.7, and 28.1 nm, respectively. Indeed, as the growth of the Sn-Ga_2_O_3_ epilayer proceeded, the evolutionary growth resulted in poor surface morphology with the protruding grains, as evidenced in Figure 5. Nevertheless, sample C2 surprisingly featured the highest crystal quality (i.e., the lowest dislocation density) among all samples due to the local competition for growth. Hence, we need to focus on the microstructures of samples C1 and C2 for a comparison.

Figure 8 displays TEM images of samples C1 (Figure 8a–c) and C2 (Figure 8d–f) when observed along the $\langle 10\bar{1}0\rangle$ axis. In Figure 8a, it is evident that a high density of dislocations generated by the sapphire substrates was blocked by an 80-nm-thick (Al_0.3_Ga_0.7_)_2_O_3_ layer, resulting in fewer dislocations in the upper Sn-Ga_2_O_3_ layer. The evolutionary growth behavior was confirmed as that of grains that became large. The reduction in dislocations aligned with an improved crystal quality, as the FWHM of the XRC decreased from 2500 to 1611 arcsec for samples A1 and C1, respectively. Sample C2 exhibited an even more dramatic reduction in dislocations, as depicted in Figure 8d, with the FWHM of the XRC further decreasing from 2282 to 1549 arcsec for samples B2 and C2, respectively. The contrast of the (Al_0.3_Ga_0.7_)_2_O_3_ layer was diminished in sample C2, indicating the diffusion of the Al content during the rapid thermal annealing. Enlarged TEM images at the interfaces of Ga_2_O_3_/sapphire (Figure 8b,e) and (Al_0.3_Ga_0.7_)_2_O_3_/Sn-Ga_2_O_3_ (Figure 8c,f) were compared for samples C1 and C2, respectively; the insets in the figure are the results of the fast Fourier transform (FFT). At the Ga_2_O_3_/sapphire interface in sample C1, a misfit dislocation with a period of 8.6 nm was observed, as shown in Figure 8b. A 4.8% difference in the lattice constant resulted in the generation of misfit dislocations and compressive strain in the grown α-Ga_2_O_3_ layer [43]. The α phase was dominant for all of the grown layers of sample C1. Hence, the topmost Sn-Ga_2_O_3_ layer also featured the α phase, as shown in the FFT in Figure 8c. At the interface of the Sn-Ga_2_O_3_/(Al_0.3_Ga_0.7_)_2_O_3_ layers, the misfit dislocation seemed to be suppressed due to the lattice mismatch reduction with the (Al_0.3_Ga_0.7_)_2_O_3_ layer. On the other hand, sample C2 had mixed features of α and κ phases compared to those of sample C1, as shown in Figure 8e. The κ phase tended to be obvious in the microstructure after thermal annealing and propagated up to the topmost Sn-Ga_2_O_3_ region, as shown in Figure 8f. For the incident [100] direction in α-Ga_2_O_3_, several diffraction spots were observed, such as on the (006), (030), (012), and (104) planes, while diffraction spots on the (004) and (060) planes were observed for κ-Ga_2_O_3_, as shown in Figure 8e,f. Then, we determined the relationship between the preferred orientations for (006) α-Ga_2_O_3_//(004) κ-Ga_2_O_3_ and [030] α-Ga_2_O_3_//[060] κ-Ga_2_O_3_. This relationship could also correspond to [1,2,3,4,5,6,7,8,9,10] α-Ga_2_O_3_//[310] κ-Ga_2_O_3_ and [110] α-Ga_2_O_3_//[1–30] κ-Ga_2_O_3_ because of the rotation of the domain of κ-Ga_2_O_3_ on sapphire by 120° [44]. The interface of Sn-Ga_2_O_3_/(Al_0.3_Ga_0.7_)_2_O_3_ was not distinguished, confirming the interfusion of Al contents and the recrystallization after rapid thermal annealing.

Figure 9 plots the average lattice spacing values obtained from the FFT patterns of the TEM analysis. For comparison, the theoretical (006) and (104) lattice spacings of α-Ga_2_O_3_ were drawn at 0.2238 and 0.2650 nm, respectively, as shown by the horizontal dotted lines in Figure 9. The measured lengths of atoms for sample C1 were similar to the theoretical values, ranging from 0.2207 to 0.2262 nm for the (006) plane and 0.2638 to 0.2678 nm for the (104) plane. As the Al content was included, the average spacings of the α-(Al_0.3_Ga_0.7_)_2_O_3_ layer ranged from 0.2187 to 0.2203 nm for the (006) plane and 0.2571 to 0.2599 nm for the (104) plane because Al was substituted in place of Ga, reducing the lattice constants of the *a*- and *c*-axes. The amount of substituted Al in sample C1 was determined to be ~27% in the FFT analysis, which was almost consistent with the composition from the XRD analysis. The lattice spacing of the Sn-Ga_2_O_3_ layer for sample C1 was slightly smaller and was partially strained from the α-(Al_0.3_Ga_0.7_)_2_O_3_ layer. In the case of sample C2, the lattice spacings of the α-Ga_2_O_3_ layer ranged from 0.2205 to 0.2244 nm for the (006) plane and from 0.2643 to 0.2650 nm for the (104) plane, and they were almost identical to those of C1. However, the lattice spacings of the α-(Al_0.3_Ga_0.7_)_2_O_3_ layer increased to 0.2206–0.2234 nm for the (006) plane and 0.2649–0.2678 nm for the (104) plane. The increase in in-plane and out-of-plane lattices implied a reduction in Al content rather than lattice distortion. In addition, the lattice of the Sn-Ga_2_O_3_ layer for sample C2 was almost identical to that of the relaxed α-Ga_2_O_3_, unlike the strained distortion of sample C1.

We conceptually schematized the growth behaviors of the Sn-Ga_2_O_3_ layer with double buffer layers in Figure 10. The (Al_0.3_Ga_0.7_)_2_O_3_/Ga_2_O_3_ double buffer layers initially included two α and κ phases, as shown in Figure 10a (sample A1). As the growth proceeded, the surface of the Sn-Ga_2_O_3_ layer became rougher. After growing the Sn-Ga_2_O_3_ layer, the κ phase was retained, as shown in Figure 10b (sample C1). The dislocation density was considerably reduced by the (Al_0.3_Ga_0.7_)_2_O_3_ blocking layer. Rapid thermal annealing of the (Al_0.3_Ga_0.7_)_2_O_3_/Ga_2_O_3_ double buffer layers resulted in the diffusion of Al to the underlying Ga_2_O_3_ and the deformation of the mosaicity on the surface, as shown in Figure 10c. The κ-phase regions became more predominant depending on the annealing temperature. When the annealing temperature was sufficiently high (e.g., >800 °C), the β phase mainly occupied the grown Sn-Ga_2_O_3_ regions. Under the growth of the topmost Sn-Ga_2_O_3_ layer, the growth competition between α- and β-phase regions resulted in local lateral growth, as shown in Figure 10d. As the β phase occupied most regions with increasing annealing temperatures, the effect of local lateral overgrowth in the Sn-Ga_2_O_3_ layer was weakened. The protruding grains might be attributed to the grain coalescence after the local lateral overgrowth. These behaviors resulted in higher surface roughness, while the crystal quality was slightly improved compared to that of the Sn-Ga_2_O_3_ layer on the non-annealed double buffer layers.

### 3.3. Electrical Properties of the Grown Sn-Doped Ga_2_O_3_ Layers

Table 2 shows the electrical properties of the grown layers after step C; they were obtained through Hall measurements at room temperature. For reference and comparison, the values of the unbuffered single Sn-doped α-Ga_2_O_3_ layer are listed in the first row of Table 2; the carrier concentration and electron mobility were recorded as 1.8 × 10^18^ cm^−3^ and 3.8 cm^2^/V·s, respectively [10]. Note that the values for samples Al–B4 are not provided because the Hall effect was not measurable on the highly resistive layers with (Al_0.3_Ga_0.7_)_2_O_3_/Ga_2_O_3_ layers. The carrier concentrations of samples C1–C3 were 1.6 × 10^19^, 4.1 × 10^18^, and 7.4 × 10^18^ cm^−3^, respectively. The variations in carrier concentration with the order were not surprising because the vicinal doping concentration and the growth rate have been previously found elsewhere in horizontal mist CVD [19,37]. Notably, several scattering factors, such as dislocations, carrier concentration, and surface roughness, can decrease mobility [45,46]. The electron mobilities of samples C1–C3 were 3.7, 21.1, and 13.0 cm^2^/V·s, respectively. The electron mobility of sample C1 seemed reasonable because of its higher carrier concentration and lower crystal quality compared to those of the non-buffered Ga_2_O_3_ layer from our previous study [10]. The relatively low crystal quality of sample C1 (despite adopting buffer layers) might be attributed to the poor interfacial quality from the two-step growth process. Hence, the increase in electron mobility in samples C2 and C3 was significant. Considering the small variations in crystal quality among samples C1–C3, the smoothed interface and the diffusion of Al through rapid thermal annealing accounted for the improvement of electron mobility. The diffusion of Al into the underlying Ga_2_O_3_ layer could cause the thickening of an Al-content-resistive layer. The carrier scattering at the poor interface could be considerably suppressed by adequate annealing temperatures of 700–750 °C. Sample C2 had the highest in-plane crystallinity and phase purity compared to the other layers with annealed double buffer layers, resulting in excellent mobility. With the use of the double buffer layers and an annealing temperature above 800 °C, the Hall effect was not measured due to the poor crystallinity resulting from the β-phase transition. Therefore, the rapid thermal annealing of (Al_x_Ga_1−x_)_2_O_3_/Ga_2_O_3_ double buffer layers was considerably effective in controlling the crystallinity of the grown Sn-Ga_2_O_3_ epilayer in a short time.

## 4. Conclusions

In conclusion, we used mist CVD to grow Sn-doped Ga_2_O_3_ epilayers with (Al_0.3_Ga_0.7_)_2_O_3_/Ga_2_O_3_ double buffer layers. The insertion of double buffer layers effectively improved the crystal quality of the upper-grown Sn-doped Ga_2_O_3_ thin films by blocking dislocations generated by the substrates. Rapid thermal annealing unintentionally resulted in surface roughening and phase mixing. At annealing temperatures of 700–750 °C, the slight mixing of the κ and β phases improved the in-plane crystal quality through local lateral overgrowth. However, a considerable phase transition to the β structure at a high annealing temperature of 800 °C impaired the crystallinity of the grown Ga_2_O_3_ layer. The highest electron mobility of the Sn-doped Ga_2_O_3_ was found in the sample annealed at 700 °C. In the case of a slightly phase-mixed Sn-doped Ga_2_O_3_ epilayer, the smoothed interface and the diffusion of Al due to rapid thermal annealing contributed to an improvement in electron mobility. Therefore, the rapidly annealed double buffer layer will be useful in shortening the processing time and achieving strong electrical properties for next-generation Ga_2_O_3_ semiconductor devices.

## Figures and Tables

**Figure 1 nanomaterials-14-00178-f001:**
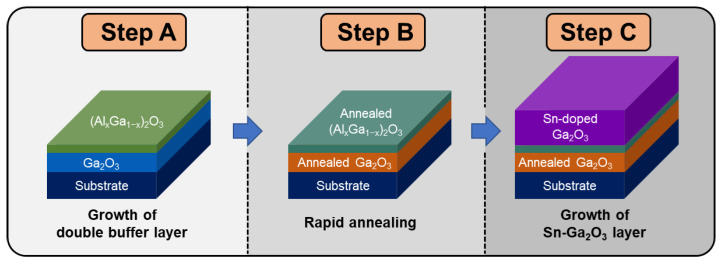
Schematic of an experimental procedure. Step A is the growth of the (Al_x_Ga_1−x_)_2_O_3_/Ga_2_O_3_ buffer layers. Step B is rapid thermal annealing. Step C is the growth of the Sn-doped α-Ga_2_O_3_ on the double buffer layers.

**Figure 2 nanomaterials-14-00178-f002:**
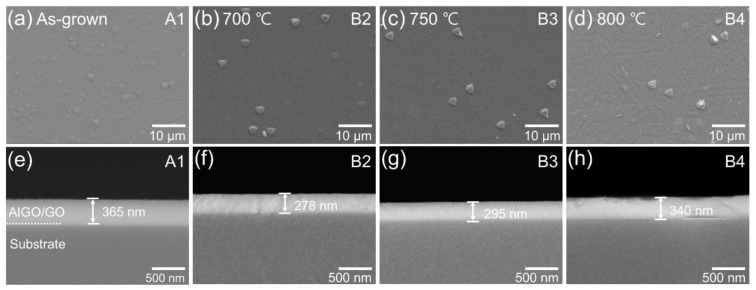
FE-SEM images of samples A1–B4: (**a**–**d**) top view and (**e**–**h**) cross-sectional view. Note that α-Ga_2_O_3_ and α-(Al_x_Ga_1−x_)_2_O_3_ are denoted by GO and AlGO, respectively.

**Figure 3 nanomaterials-14-00178-f003:**
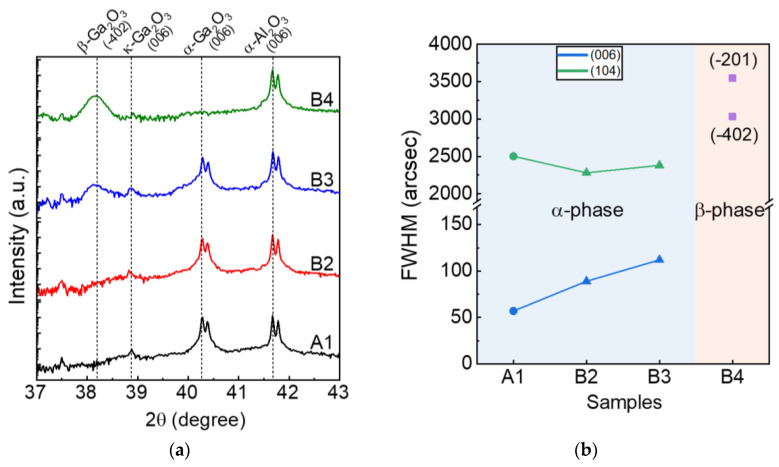
(**a**) XRD θ/2θ scan of the grown α-(Al_x_Ga_1−x_)_2_O_3_/α-Ga_2_O_3_. The α-Al_2_O_3_ (006) and κ-Ga_2_O_3_ (006) peaks were found in all samples at 41.67° and 38.9°, respectively. The α-Ga_2_O_3_ (006) peak was observed at 40.25° (samples A1–B3). The β-Ga_2_O_3_ (−402) peaks were observed at 38.3° above the annealing temperature of 750 °C (samples B3 and B4). Note that double peaks originated from an X-ray source (Cu) with Kα1 and Kα2. (**b**) The FWHM of the XRD ω scan for samples A1–B4.

**Figure 4 nanomaterials-14-00178-f004:**
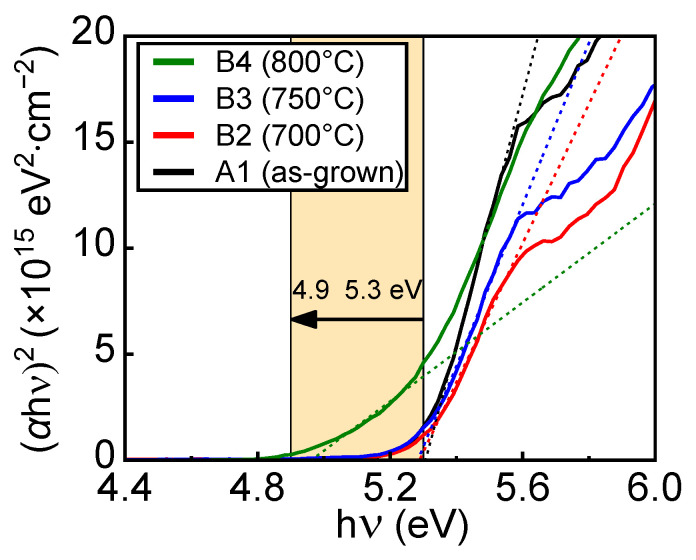
Tauc plot of measured transmittance for samples A1–B4. The bandgaps of samples A1, B2, and B3 were similarly ~5.3 eV, indicating that the α phase was dominant. The bandgap of sample B4 was 4.9 eV, corresponding to the typical β phase.

**Figure 5 nanomaterials-14-00178-f005:**
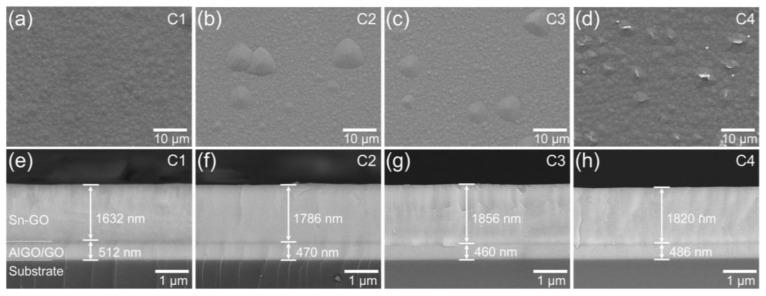
FE-SEM images of samples C1–C4: (**a**–**d**) top view and (**e**–**h**) cross-sectional view. Note that Ga_2_O_3_ and (Al_x_Ga_1−x_)_2_O_3_ are denoted as GO and AlGO, respectively.

**Figure 6 nanomaterials-14-00178-f006:**
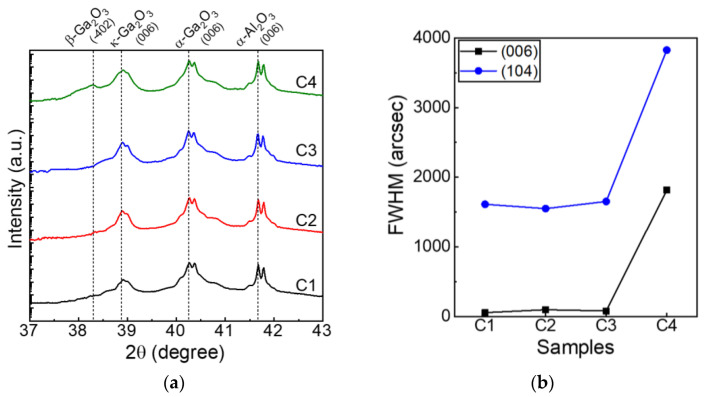
(**a**) XRD θ/2θ scan of the grown Sn-doped α-Ga_2_O_3_ epilayers. The α-Al_2_O_3_ (006) and κ-Ga_2_O_3_ (006) peaks were found in all samples (C1–C4) at 41.67° and 38.9°, respectively. The β-Ga_2_O_3_ −402) peaks were observed at 38.3° with an annealing temperature of 800 °C (sample C4). (**b**) The FWHM of the XRD ω scan for samples C1–C4.

**Figure 7 nanomaterials-14-00178-f007:**
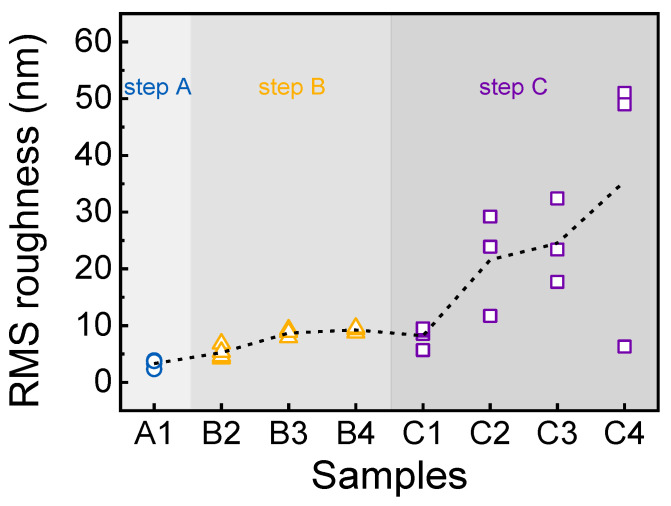
RMS roughness of samples A1–C4 measured with AFM. The dotted line indicates the average value.

**Figure 8 nanomaterials-14-00178-f008:**
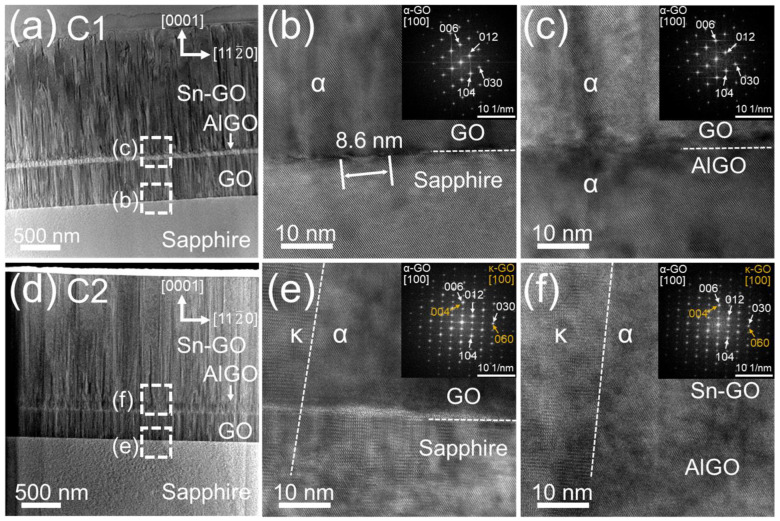
TEM images: (**a**,**d**) whole cross-section, (**b**,**e**) sapphire/α-Ga_2_O_3_, and (**c**,**f**) (Al_x_Ga_1−x_)_2_O_3_/Sn-Ga_2_O_3_ regions. The insets are the FFT patterns of the corresponding regions. Note that Ga_2_O_3_ and (Al_x_Ga_1−x_)_2_O_3_ are denoted as GO and AlGO, respectively.

**Figure 9 nanomaterials-14-00178-f009:**
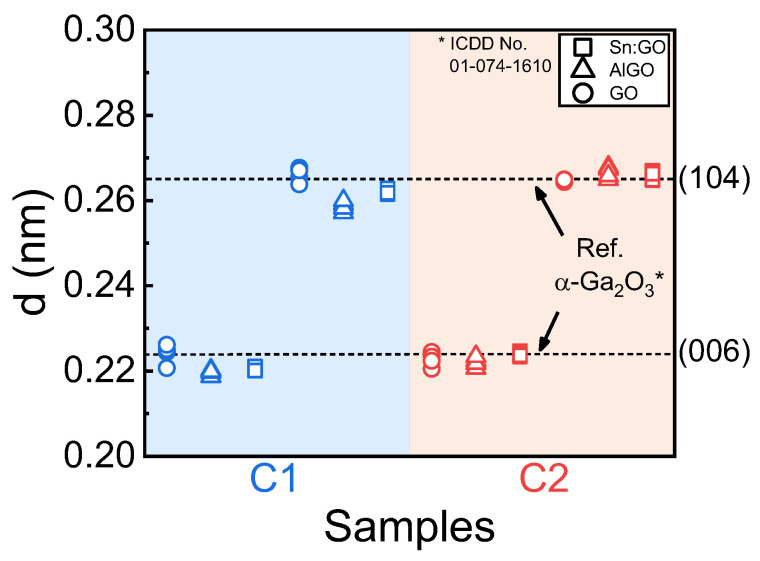
Comparison of the lattice spacing for each layer of samples C1 and C2 obtained from the FFT patterns of the TEM images. The dotted lines indicate the theoretical values of the α-Ga_2_O_3_ layer on the (006) and (104) planes.

**Figure 10 nanomaterials-14-00178-f010:**
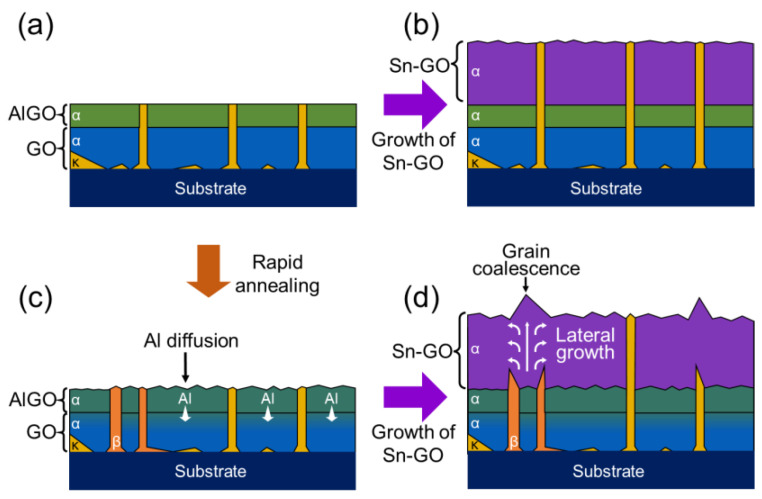
Schematic of the growth mechanism for the Ga_2_O_3_ layer with the double buffer layers. (**a**) As-grown (Al_x_Ga_1−x_)_2_O_3_/Ga_2_O_3_ double buffers; (**b**) Sn-Ga_2_O_3_ layer on the as-grown (Al_x_Ga_1−x_)_2_O_3_/Ga_2_O_3_ double buffers; (**c**) thermally annealed (Al_x_Ga_1−x_)_2_O_3_/Ga_2_O_3_ double buffers; (**d**) Sn-Ga_2_O_3_ layer on the thermally annealed (Al_x_Ga_1−x_)_2_O_3_/Ga_2_O_3_ double buffers. Note that Ga_2_O_3_ and (Al_x_Ga_1−x_)_2_O_3_ are denoted as GO and AlGO, respectively.

**Table 1 nanomaterials-14-00178-t001:** Summary of the experimental procedures.

Step		Step A	Step B			Step C
Sample		A1	B2	B3	B4	C1–C4
Process		Buffer growth	Heat-treat	Heat-treat	Heat-treat	Film Growth
Template		*c*-sapphire	A1	A1	A1	A1–B4
Temperature		450/500 °C	700 °C	750 °C	800 °C	450 °C
Process time		1/2 h	3 min	3 min	3 min	3 h
Precursor Conc.	Ga(acac)_3_	0.03 M/0.05 M				
	Al(acac)_3_	0.18 M/—				
	SnCl_2_					0.1 mol%
Gas	Carrier (air)	5 L/min	N_2_	N_2_	N_2_	5 L/min
	Dilution (O_2_)	5 L/min				5 L/min

**Table 2 nanomaterials-14-00178-t002:** Comparison of electrical properties for a non-buffer layer and samples A1–C4.

Sample	Process	Carrier Concentration (cm^−3^)	Electron Mobility (cm^2^/V·s)
Ref. [19]	No buffer	1.8 × 10^18^	3.8
C1	Buffers (no annealing) Buffers (annealing at 700 °C) Buffers (annealing at 750 °C) Buffers (annealing at 800 °C)	1.6 × 10^19^	3.7
C2		4.1 × 10^18^	21.1
C3		7.4 × 10^18^	13.0
C4		—	—

## Data Availability

Data are contained within the article.

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
