# Peer review of "Controlled Crystallinity of a Sn-Doped α-Ga2O3 Epilayer Using Rapidly Annealed Double Buffer Layers"

_nanomaterials, 2024, doi:10.3390/nano14020178_

Round 1
Reviewer 1 Report
Comments and Suggestions for Authors
In this manuscript entitled “Controlled Crystallinity of Sn-Doped α-Ga2O3 Epilayer Using Rapidly Annealed Double Buffer Layers”, Kim et al. have systematically explored the effect of the double buffer layers composed of (AlxGa1-x)2O3/Ga2O3 on the mist chemical vapor deposition growth of the Sn doped α-Ga2O3 nanofilms. It has been determined that the insertion of the (AlxGa1-x)2O3/Ga2O3 double buffer layers improved the crystal quality by blocking the dislocation generated from the substrates. Moreover, the electron mobility of the Sn doped α-Ga2O3 thin films has been considerably improved due to the smoothened interface and the Al diffusion. Overall, wide bandgap semiconductor is a hot research topic with broad research interest. This study provides an efficient route to improve the quality of the representative wide-bandgap semiconductor α-Ga2O3. However, in the current stage, it is found that there is still room for further improvement. Therefore, a major revision is recommended. The following comments should be fully taken into account during your revision.
1. Some experimental details have been missed. For example, “a 0.05-M Ga precursor solution and a 0.1% SnCl2 solution were mixed” (Page 2), what’s the volume ratio here used for the mixing?
2. Page 3, Figure 1, Step B, “(AlxGa1-x)2O3” is suggested to be changed to “annealed (AlxGa1-x)2O3”.
3. Why is (AlxGa1-x)2O3 chosen as the buffer layer? Basically, Al has a smaller diameter than Ga. Therefore, doping Al into Ga2O3 will result in the decreased lattice constant. By contrast, Sn has a larger diameter than Ga. Therefore, doping Sn into Ga2O3 will result in the enlarged lattice constant. In this consideration, the lattice mismatch will increase by using (AlxGa1-x)2O3 as the buffer layer. So what’s your starting point for choosing it as the buffer layer?
4. Page 5, it is said that “band gaps of samples B2 and B3 were 5.29 and 5.20 eV”. But I find that both values are actually very close to ≈5.3 eV (Page 6, Figure 4). The authors should have a check on the description.
5. Page 8, Figure 8, the SAED patterns lack the scale bars. They should be provided. Otherwise, the figures have no significance.
6. In the first sentence of “Gallium oxide (Ga2O3) is a next-generation high-voltage power semiconductor material that features a wide band gap…”, the latest developments in various aspects (e.g., Materials 2022, 15, 913; J. Mater. Sci. Technol. 2023, 163, 150–157; Rare Met. 2022, 41, 1375–1379) should be included to enrich the manuscript.
7. There are some grammatical errors/typos. For example, Page 4, should “Figures 2e-g” be “Figures 2e-h”? In addition, the hyphen sign ‘-’ is widely mistakenly used as the minus sign (−). The whole manuscript should be carefully checked.
Author Response
Please check our response with attachment. Thank you very much.

Reviewer 2 Report
Comments and Suggestions for Authors
The paper details the growth of Sn-doped α-Ga2O3 epilayers using mist chemical vapor deposition (CVD) with double buffer layers. It emphasizes improving epilayer crystallinity and reducing dislocations. The study explores the effects of rapid thermal annealing on these layers. Results demonstrate enhanced electron mobility and crystallinity. The findings contribute to understanding Ga2O3's potential in high-power electronic devices, addressing key challenges in material quality for semiconductor applications. This research is significant in advancing the development of high-performance semiconductor materials. The paper is meticulously researched and carefully discussed. The quality of the measurement data is outstanding, and the content is summarized at a very high level. I have determined that this paper's content is sufficient for publication in "Nanomaterials."
Author Response
We express our deep gratitude to Reviewer for generous and good comments. We hope this paper will be helpful for the potential readers of Nanomaterials.
Reviewer 3 Report
Comments and Suggestions for Authors
Sn doping is an effective way to improve the opto-electronic properties of Ga2O3 thin films. In this paper, a Sn-doped α-Ga2O3 epilayer was successfully grown with high crystallinity and electron mobility. To do so, the authors controlled the crystallinity of α-Ga2O3 thin films grown by mist CVD with a combination process of double buffer layers and rapid thermal annealing. The double buffer layers consist of an α-Ga2O3 and an α-(AlxGa1-x)2O3 layers for the strain relaxation of the heteroepitaxial interface. The temperature of rapid thermal annealing was adjusted to control the phase transition and interfacial quality of the grown layers over a short time. The films were characterized by X-ray diffraction, field-emission scanning electron microscopy, transmission electron microscopy for microstructural characterizations. Ultraviolet visible spectroscopy for optical properties, and atomic force microscope for surface morphology. The electrical properties, carrier concentration and mobility were determined using Hall effect measurement. Over all this work provides important method and knowledge to guide the fabrication of Sn-doped Ga2O3 thin films
for optoelectronic applications. Following are few remarks
1) The XRD pattern, θ/2θ scan of the grown α-(AlxGa1-x)2O3/α-Ga2O3, shown in Figure 3a shows that all peaks are doubled. Please assign all the peaks observed in the pattern and provide more details in caption figure.
2) Generally the figure captions, for instance Figure 4 and 6, are not enough in detail to describe the figures clear.
Author Response

(The authors gave the same response as above.)

Reviewer 4 Report
Comments and Suggestions for Authors
The authors reported on the controlled the crystallinity of Sn-doped α-Ga2O3 epilayer using rapidly annealed double buffer layers
The manuscript discussed clearly on the α-phase Ga2O3 growth with the buffer layer, which is quite interesting. The growth approach is also viable to achieve better electrical performance of the α-Ga2O3 epilayer layer.
The presented XRD, TEM results on the crystallinity are clearer to support the growth of the Sn-doping α-Ga2O3 epilayer
I recommend acceptance of the manuscript in present form; no further revision is required.
Author Response
We really appreciate Reviewer for positive comments. We hope this paper will be helpful for the potential readers of Nanomaterials.
Round 2
Reviewer 1 Report
Comments and Suggestions for Authors
It is glad to find that the previous suggestions have been well addressed by the authors. Now the manuscript can be recommended for publication.